# CogView: Mastering Text-to-Image Generation via Transformers

**Ming Ding**[†], **Zhuoyi Yang**[†], **Wenyi Hong**[†], **Wendi Zheng**[†], **Chang Zhou**[‡], **Da Yin**[†],
**Junyang Lin**[‡], **Xu Zou**[†], **Zhou Shao**[♠], **Hongxia Yang**[‡], **Jie Tang**[†♠]
[†]Tsinghua University  [‡]DAMO Academy, Alibaba Group  [♠]BAAI
{dm18@mails, jietang@mail}.tsinghua.edu.cn

## Abstract

Text-to-Image generation in the general domain has long been an open problem, which requires both a powerful generative model and cross-modal understanding. We propose CogView, a 4-billion-parameter Transformer with VQ-VAE tokenizer to advance this problem. We also demonstrate the finetuning strategies for various downstream tasks, e.g. style learning, super-resolution, text-image ranking and fashion design, and methods to stabilize pretraining, e.g. eliminating NaN losses. CogView achieves the state-of-the-art FID on the blurred MS COCO dataset, outperforming previous GAN-based models and a recent similar work DALL-E. [1]

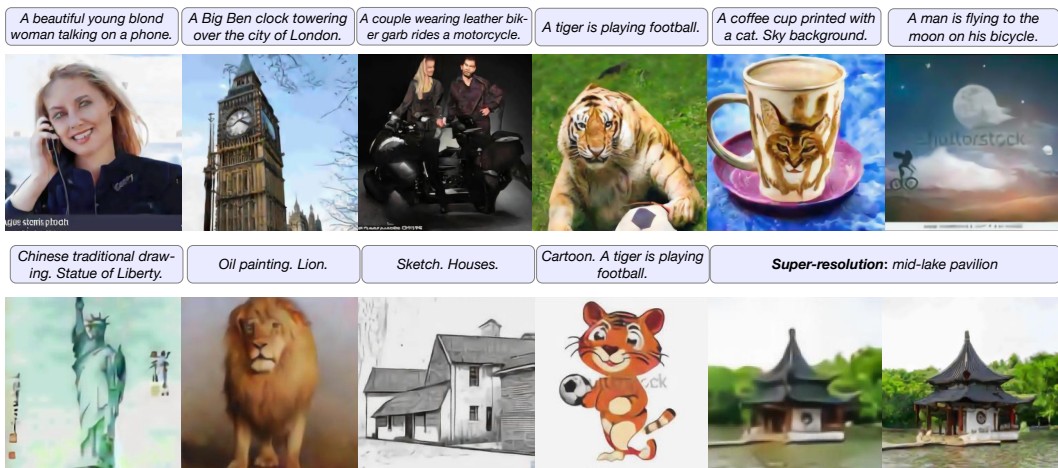

Figure 1: Samples generated by CogView. The text in the first line is either from MS COCO (outside our training set) or user queries on our demo website. The images in the second line are finetuned results for different styles or super-resolution. The actual input text is in Chinese, which is translated into English here for better understanding. More samples for captions from MS COCO are included in Appendix F.

## 1 Introduction

> *"There are two things for a painter, the eye and the mind... eyes, through which we view the nature; brain, in which we organize sensations by logic for meaningful expression." (Paul Cézanne [17])*

---

[1]Codes and models are at https://github.com/THUDM/CogView. We also have a demo website of our latest model at https://wudao.aminer.cn/CogView/index.html (without post-selection).

35th Conference on Neural Information Processing Systems (NeurIPS 2021).

As contrastive self-supervised pretraining has revolutionized computer vision (CV) [24, 21, 8, 32], visual-language pretraining, which brings high-level semantics to images, is becoming the next frontier of visual understanding [38, 30, 39]. Among various pretext tasks, text-to-image generation expects the model to (1) disentangle shape, color, gesture and other features from pixels, (2) understand the input text, (2) align objects and features with corresponding words and their synonyms and (4) learn complex distributions to generate the overlapping and composite of different objects and features, which, like painting, is beyond basic visual functions (related to eyes and the V1–V4 in brain [22]), requiring a higher-level cognitive ability (more related to the angular gyrus in brain [3]).

The attempts to teach machines text-to-image generation can be traced to the early times of deep generative models, when Mansimov et al. [35] added text information to DRAW [20]. Then Generative Adversarial Nets [19] (GANs) began to dominate this task. Reed et al. [42] fed the text embeddings to both generator and discriminator as extra inputs. StackGAN [54] decomposed the generation into a sketch-refinement process. AttnGAN [51] used attention on words to focus on the corresponding subregion. ObjectGAN [29] generated images following a text→boxes→layouts→image process. DM-GAN [55] and DF-GAN [45] introduced new architectures, e.g. dyanmic memory or deep fusion block, for better image refinement. Although these GAN-based models can perform reasonable synthesis in simple and domain-specific dataset, e.g. Caltech-UCSD Birds 200 (CUB), the results on complex and domain-general scenes, e.g. MS COCO [31], are far from satisfactory.

Recent years have seen a rise of the auto-regressive generative models. Generative Pre-Training (GPT) models [37, 4] leveraged Transformers [48] to learn language models in large-scale corpus, greatly promoting the performance of natural language generation and few-shot language understanding [33]. Auto-regressive model is not nascent in CV. PixelCNN, PixelRNN [47] and Image Transformer [36] factorized the probability density function on an image over its sub-pixels (color channels in a pixel) with different network backbones, showing promising results. However, a real image usually comprises millions of sub-pixels, indicating an unaffordable amount of computation for large models. Even the biggest pixel-level auto-regressive model, ImageGPT [7], was pretrained on ImageNet at a max resolution of only $96 \times 96$.

The framework of Vector Quantized Variational AutoEncoders (VQ-VAE) [46] alleviates this problem. VQ-VAE trains an encoder to compress the image into a low-dimensional discrete latent space, and a decoder to recover the image from the hidden variable in the stage 1. Then in the stage 2, an auto-regressive model (such as PixelCNN [47]) learns to fit the prior of hidden variables. This discrete compression loses less fidelity than direct downsampling, meanwhile maintains the spatial relevance of pixels. Therefore, VQ-VAE revitalized the auto-regressive models in CV [41]. Following this framework, Esser et al. [15] used Transformer to fit the prior and further switches from $L_2$ loss to GAN loss for the decoder training, greatly improving the performance of domain-specific unconditional generation.

The idea of CogView comes naturally: large-scale generative joint pretraining for both text and image (from VQ-VAE) tokens. We collect 30 million high-quality (Chinese) text-image pairs and pretrain a Transformer with 4 billion parameters. However, large-scale text-to-image generative pretraining could be very unstable due to the heterogeneity of data. We systematically analyze the reasons and solved this problem by the proposed *Precision Bottleneck Relaxation* and *Sandwich Layernorm*. As a result, CogView greatly advances the quality of text-to-image generation.

A recent work DALL-E [39] independently proposed the same idea, and was released earlier than CogView. Compared with DALL-E, CogView steps forward on the following four aspects:

- CogView outperforms DALL-E and previous GAN-based methods at a large margin according to the Fréchet Inception Distance (FID) [25] on blurred MS COCO, and is the first open-source large text-to-image transformer.

- Beyond zero-shot generation, we further investigate the potential of finetuning the pretrained CogView. CogView can be adapted for diverse downstream tasks, such as style learning (domain-specific text-to-image), super-resolution (image-to-image), image captioning (image-to-text), and even text-image reranking.

- The finetuned CogView enables self-reranking for post-selection, and gets rid of an additional CLIP model [38] in DALL-E. It also provides a new metric *Caption Loss* to measure the quality and accuracy for text-image generation at a finer granularity than FID and Inception Score (IS) [43].

- We proposed PB-relaxation and Sandwich-LN to stabilize the training of large Transformers on complex datasets. These techniques are very simple and can eliminate overflow in forwarding (characterized as NaN losses), and make CogView able to be trained with *almost FP16* (O2[2]). They can also be generalized to the training of other transformers.

## 2 Method

### 2.1 Theory

In this section, we will derive the theory of CogView from VAE[3] [26]: *CogView optimizes the Evidence Lower BOund (ELBO) of joint likelihood of image and text.* The following derivation will turn into a clear re-interpretation of VQ-VAE if without text $\mathbf{t}$.

Suppose the dataset $(\mathbf{X}, \mathbf{T}) = \{x_i, t_i\}_{i=1}^{N}$ consists of $N$ i.i.d. samples of image variable $\mathbf{x}$ and its description text variable $\mathbf{t}$. We assume the image $\mathbf{x}$ can be generated by a random process involving a latent variable $\mathbf{z}$: (1) $t_i$ is first generated from a prior $p(\mathbf{t}; \theta)$. (2) $z_i$ is then generated from the conditional distribution $p(\mathbf{z}|\mathbf{t} = t_i; \theta)$. (3) $x_i$ is finally generated from $p(\mathbf{x}|\mathbf{z} = z_i; \psi)$. We will use a shorthand form like $p(x_i)$ to refer to $p(\mathbf{x} = x_i)$ in the following part.

Let $q(\mathbf{z}|x_i; \phi)$ be the variational distribution, which is the output of the encoder $\phi$ of VAE. The log-likelihood and the evidence lower bound (ELBO) can be written as:

$$\log p(\mathbf{X}, \mathbf{T}; \theta, \psi) = \sum_{i=1}^{N} \log p(t_i; \theta) + \sum_{i=1}^{N} \log p(x_i|t_i; \theta, \psi) \tag{1}$$

$$\geq -\sum_{i=1}^{N} \left( \underbrace{-\log p(t_i; \theta)}_{\text{NLL loss for text}} + \underbrace{\mathbb{E}_{z_i \sim q(\mathbf{z}|x_i;\phi)}[-\log p(x_i|z_i; \psi)]}_{\text{reconstruction loss}} + \underbrace{\text{KL}\big(q(\mathbf{z}|x_i;\phi)\|p(\mathbf{z}|t_i;\theta)\big)}_{\text{KL between } q \text{ and (text conditional) prior}} \right). \tag{2}$$

The framework of VQ-VAE differs with traditional VAE mainly in the KL term. Traditional VAE fixes the prior $p(\mathbf{z}|t_i; \theta)$, usually as $\mathcal{N}(0, \mathbf{I})$, and learns the encoder $\phi$. However, it leads to *posterior collapse* [23], meaning that $q(\mathbf{z}|x_i; \phi)$ sometimes collapses towards the prior. VQ-VAE turns to fix $\phi$ and fit the prior $p(\mathbf{z}|t_i; \theta)$ with another model parameterized by $\theta$. This technique eliminates posterior collapse, because the encoder $\phi$ is now only updated for the optimization of the reconstruction loss. In exchange, the approximated posterior $q(\mathbf{z}|x_i; \phi)$ could be very different for different $x_i$, so we need a very powerful model for $p(\mathbf{z}|t_i; \theta)$ to minimize the KL term.

Currently, the most powerful generative model, Transformer (GPT), copes with sequences of tokens over a discrete codebook. To use it, we make $\mathbf{z} \in \{0, ..., |V| - 1\}^{h \times w}$, where $|V|$ is the size of codebook and $h \times w$ is the number of dimensions of $\mathbf{z}$. The sequences $z_i$ can be either sampled from $q(\mathbf{z}|x_i; \phi)$, or directly $z_i = \text{argmax}_{\mathbf{z}} \, q(\mathbf{z}|x_i; \phi)$. We choose the latter for simplicity, so that $q(\mathbf{z}|x_i; \phi)$ becomes a one-point distribution on $z_i$. The Equation (2) can be rewritten as:

$$-\sum_{i=1}^{N} \left( \underbrace{\mathbb{E}_{z_i \sim q(\mathbf{z}|x_i;\phi)}[-\log p(x_i|z_i; \psi)]}_{\text{reconstruction loss}} \underbrace{-\log p(t_i; \theta)}_{\text{NLL loss for text}} \underbrace{-\log p(z_i|t_i; \theta)}_{\text{NLL loss for } \mathbf{z}} \right). \tag{3}$$

The learning process is then divided into two stages: (1) The encoder $\phi$ and decoder $\psi$ learn to minimize the reconstruction loss. (2) A single GPT optimizes the two negative log-likelihood (NLL) losses by concatenating text $t_i$ and $z_i$ as an input sequence.

As a result, the first stage degenerates into a pure discrete Auto-Encoder, serving as an *image tokenizer* to transform an image to a sequence of tokens; the GPT in the second stage undertakes most of the modeling task. Figure 3 illustrates the framework of CogView.

---

[2]meaning that all computation, including forwarding and backwarding are in FP16 without any conversion, but the optimizer states and the master weights are FP32.

[3]In this paper, **bold** font denotes a random variable, and regular font denotes a concrete value. See this comprehensive tutorial [12] for the basics of VAE.

## 2.2 Tokenization

In this section, we will introduce the details about the tokenizers in CogView and a comparison about different training strategies about the image tokenizer (VQVAE stage 1).

Tokenization for text is already well-studied, e.g. BPE [16] and SentencePiece [28]. In CogView, we ran SentencePiece on a large Chinese corpus to extract 50,000 text tokens.

The image tokenizer is a discrete Auto-Encoder, which is similar to the stage 1 of VQ-VAE [46] or d-VAE [39]. More specifically, the Encoder $\phi$ maps an image $x$ of shape $H \times W \times 3$ into $\text{Enc}_\phi(x)$ of shape $h \times w \times d$, and then each $d-$dimensional vector is quantized to a *nearby* embedding in a learnable codebook $\{v_0, ..., v_{|V|-1}\}, \forall v_k \in \mathbb{R}^d$. The quantized result can be represented by $h \times w$ indices of embeddings, and then we get the latent variable $\mathbf{z} \in \{0, ..., |V|-1\}^{h \times w}$. The Decoder $\psi$ maps the quantized vectors back to a (blurred) image to reconstruct the input. In our 4B-parameter CogView, $|V| = 8192, d = 256, H = W = 256, h = w = 32$.

The training of the image tokenizer is non-trivial due to the existence of discrete selection. Here we introduce four methods to train an image tokenizer.

- *The nearest-neighbor mapping, straight-through estimator* [2], which is proposed by the original VQVAE. A common concern of this method [39] is that, when the codebook is large and not initialized carefully, only a few of embeddings will be used due to the curse of dimensionality. We did not observe this phenomenon in the experiments.

- *Gumbel sampling, straight-through estimator*. If we follow the original VAE to reparameterize a categorical distribution of latent variable $\mathbf{z}$ based on distance between vectors, i.e. $p(\mathbf{z}_{i \times w+j} = v_k | x) = \frac{e^{-\|v_k - \text{Enc}_\phi(x)_{ij}\|_2 / \tau}}{\sum_{k=0}^{|V|-1} e^{-\|v_k - \text{Enc}_\phi(x)_{ij}\|_2 / \tau}}$, an unbiased sampling strategy is $z_{i \times w+j} = \text{argmax}_k g_k - \|v_k - \text{Enc}_\phi(x)_{ij}\|_2 / \tau, \; g_k \sim \text{Gumbel}(0,1)$, where the temperature $\tau$ is gradually decreased to 0. We can further use the differentiable softmax to approximate the one-hot distribution from argmax. DALL-E adopts this method with many other tricks to stabilize the training.

- *The nearest-neighbor mapping, moving average*, where each embedding in the codebook is updated periodically during training as the mean of the vectors recently mapped to it [46].

- *The nearest-neighbor mapping, fixed codebook*, where the codebook is fixed after initialized.

**Comparison.** To compare the methods, we train four image tokenizers with the same architecture on the same dataset and random seed, and demonstrate the loss curves in Figure 2. We find that all the methods are basically evenly matched, meaning that the learning of the embeddings in the codebook is not very important, if initialized properly. In pretraining, we use the tokenizer of moving average method.

The introduction of **data** and more details about tokenization are in Appendix A.

Figure 2: $L_2$ loss curves during training image tokenizers. All the above methods finally converge to a similar loss level.

## 2.3 Auto-regressive Transformer

The backbone of CogView is a unidirectional Transformer (GPT). The Transformer has 48 layers, with the hidden size of 2560, 40 attention heads and 4 billion parameters in total. As shown in Figure 3, four seperator tokens, [ROI1] (reference text of image), [BASE], [BOI1] (beginning of image), [EOI1] (end of image) are added to each sequence to indicate the boundaries of text and image. All the sequences are clipped or padded to a length of 1088.

The pretext task of pretraining is left-to-right token prediction, a.k.a. language modeling. Both image and text tokens are equally treated. DALL-E [39] suggests to lower the loss weight of text tokens; on the contrary, during small-scale experiments we surprisingly find the text modeling is the key for the success of text-to-image pretraining. If the loss weight of text tokens is set to zero, the model will fail to find the connections between text and image and generate images totally unrelated to the input text.

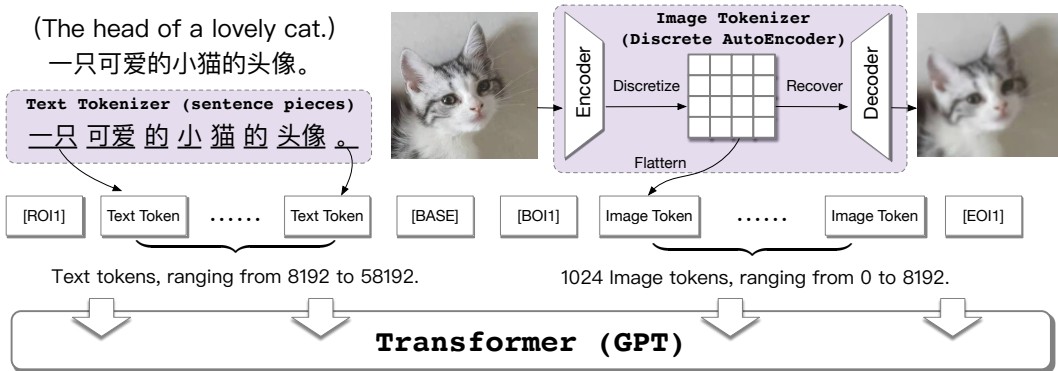

Figure 3: The framework of CogView. `[ROI1]`, `[BASE1]`, etc., are seperator tokens.

We hypothesize that text modeling abstracts knowledge in hidden layers, which can be efficiently exploited during the later image modeling.

We train the model with batch size of 6,144 sequences (6.7 million tokens per batch) for 144,000 steps on 512 V100 GPUs (32GB). The parameters are updated by Adam with max $lr = 3 \times 10^{-4}, \beta_1 = 0.9, \beta_2 = 0.95$, weight decay $= 4 \times 10^{-2}$. The learning rate warms up during the first 2% steps and decays with cosine annealing [34]. With hyperparameters in an appropriate range, we find that the training loss mainly depends on the total number of trained tokens (tokens per batch $\times$ steps), which means that doubling the batch size (and learning rate) results in a very similar loss if the same number of tokens are trained. Thus, we use a relatively large batch size to improve the parallelism and reduce the percentage of time for communication. We also design a three-region sparse attention to speed up training and save memory without hurting the performance, which is introduced in Appendix B.

## 2.4 Stabilization of training

Currently, pretraining large models (>2B parameters) usually relies on 16-bit precision to save GPU memory and speed up the computation. Many frameworks, e.g. DeepSpeed ZeRO [40], even only support FP16 parameters. However, text-to-image pretraining is very unstable under 16-bit precision. Training a 4B ordinary pre-LN Transformer will quickly result in NaN loss within 1,000 iterations. To stabilize the training is the most challenging part of CogView, which is well-aligned with DALL-E.

We summarize the solution of DALL-E as to *tolerate* the numerical problem of training. Since the values and gradients vary dramatically in scale in different layers, they propose a new mixed-precision framework *per-resblock loss scaling* and store all gains, biases, embeddings, and unembeddings in 32-bit precision, with 32-bit gradients. This solution is complex, consuming extra time and memory and not supported by most current training frameworks.

CogView instead *regularizes* the values. We find that there are two kinds of instability: overflow (characterized by NaN losses) and underflow (characterized by diverging loss). The following techniques are proposed to solve them.

**Precision Bottleneck Relaxation (PB-Relax).** After analyzing the dynamics of training, we find that overflow always happens at two *bottleneck* operations, the final LayerNorm or attention.

- In the deep layers, the values of the outputs could *explode* to be as large as $10^4 \sim 10^5$, making the variation in LayerNorm overflow. Luckily, as $\text{LayerNorm}(x) = \text{LayerNorm}(x/\max(x))$, we can relax this bottleneck by dividing the maximum first[4].

- The attention scores $Q^T K/\sqrt{d}$ could be significantly larger than input elements, and result in overflow. Changing the computational order into $Q^T(K/\sqrt{d})$ alleviates the problem. To eliminate the overflow, we notice that $\text{softmax}(Q^T K/\sqrt{d}) = \text{softmax}(Q^T K/\sqrt{d} -$

---

[4]We cannot directly divide $x$ by a large constant, which will lead to underflow in the early stage of training.

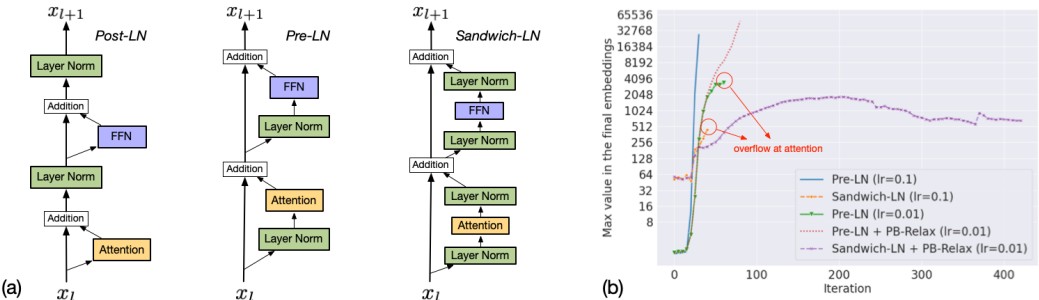

Figure 4: (a) Illustration of different LayerNorm structures in Transformers. Post-LN is from the original paper; Pre-LN is the most popular structure currently; Sandwich-LN is our proposed structure to stabilize training. (b) The numerical scales in our toy experiments with 64 layers and a large learning rate. Trainings without Sandwich-LN overflow in main branch; trainings without PB-relax overflow in attention; Only the training with both can continue.

constant), meaning that we can change the computation of attention into

$$\text{softmax}(\frac{Q^T K}{\sqrt{d}}) = \text{softmax}\left( \left( \frac{Q^T}{\alpha\sqrt{d}}K - \max(\frac{Q^T}{\alpha\sqrt{d}}K) \right) \times \alpha \right), \qquad (4)$$

where $\alpha$ is a big number, e.g. $\alpha = 32$.[5] In this way, the maximum (absolute value) of attention scores are also divided by $\alpha$ to prevent it from overflow. A detailed analysis about the attention in CogView is in Appendix C.

**Sandwich LayerNorm (Sandwich-LN).** The LayerNorms [1] in Transformers are essential for stable training. Pre-LN [50] is proven to converge faster and more stable than the original Post-LN, and becomes the default structure of Transformer layers in recent works. However, it is not enough for text-to-image pretraining. The output of LayerNorm $\frac{(x-\bar{x})\sqrt{d}}{\sqrt{\sum_i (x_i - \bar{x})^2}}\gamma + \beta$ is basically proportional to the square root of the hidden size of $x$, which is $\sqrt{d} = \sqrt{2560} \approx 50$ in CogView. If input values in some dimensions are obviously larger than the others – which is true for Transformers – output values in these dimensions will also be large ($10^1 \sim 10^2$). In the residual branch, these large values are magnified and be added back to the main branch, which aggravates this phenomenon in the next layer, and finally causes the *value explosion* in the deep layers.

This reason behind value explosion inspires us to restrict the layer-by-layer aggravation. We propose Sandwich LayerNorm, which also adds a LayerNorm at the end of each residual branch. Sandwich-LN ensures the scale of input values in each layer within a reasonable range, and experiments on training 500M model shows that its influence on convergence is negligible. Figure 4(a) illustrates different LayerNorm structures in Transformers.

**Toy Experiments.** Figure 4(b) shows the effectiveness of PB-relax and Sandwich-LN with a toy experimental setting, since training many large models for verification is not realistic. We find that *deep transformers* (64 layers, 1024 hidden size), *large learning rates* (0.1 or 0.01), *small batch size* (4) can simulate the value explosion in training with reasonable hyperparameters. PB-relax + Sandwich-LN can even stabilize the toy experiments.

**Shrink embedding gradient.** Although we did not observe any sign of underflow after using Sandwich-LN, we find that the gradient of token embeddings is much larger than that of the other parameters, so that simply shrinking its scale by $\alpha = 0.1$ increases the dynamic loss scale to further prevent underflow, which can be implemented by `emb=emb*alpha+emb.detach()*(1-alpha)` in Pytorch. It seems to slow down the updating of token embeddings, but actually does not hurt performance in our experiments, which also corresponds to a recent work MoCo v3 [9].

**Discussion.** The PB-relax and Sandwich-LN successfully stabilize the training of CogView and a 8.3B-parameter CogView-large. They are also general for all Transformer pretraining, and will enable the training of very deep Transformers in the future. As an evidence, we used PB-relax successfully eliminating the overflow in training a 10B-parameter GLM [14]. However, in general,

---

[5]The max must be at least head-wise, because the values vary greatly in different heads.

the precision problems in language pretraining is not so significant as in text-to-image pretraining. We hypothesize that the root is the heterogeneity of data, because we observed that text and image tokens are distinguished by scale in some hidden states. Another possible reason is hard-to-find underflow, guessed by DALL-E. A thorough investigation is left for future work.

## 3 Finetuning

CogView steps further than DALL-E on finetuning. Especially, we can improve the text-to-image generation via finetuning CogView for super-resolution and self-reranking. All the finetuning tasks can be completed within one day on a single DGX-2.

### 3.1 Super-resolution

Since the image tokenizer compresses $256 \times 256$-pixel images into $32 \times 32$-token sequences before training, the generated images are blurrier than real images due to the lossy compression. However, enlarging the sequence length will consume much more computation and memory due to the $O(n^2)$ complex of attention operations. Previous works [13] about super-resolution, or image restoration, usually deal with images already in high resolution, mapping the blurred local textures to clear ones. They cannot be applied to our case, where we need to add meaningful details to the generated low-resolution images. Figure 5 (b) is an example of our finetuning method, and illustrates our desired behavior of super-resolution.

The motivation of our finetuning solution for super-resolution is a belief that *CogView is trained on the most complex distribution in general domain, and the objects of different resolution has already been covered*.[6] Therefore, finetuning CogView for super-resolution should not be hard.

Specifically, we first finetune CogView into a conditional super-resolution model from $16 \times 16$ image tokens to $32 \times 32$ tokens. Then we magnify an image of $32 \times 32$ tokens to $64 \times 64$ tokens ($512 \times 512$ pixels) patch-by-patch via a center-continuous sliding-window strategy in Figure 5 (a). This order performs better that the raster-scan order in preserving the completeness of the central area.

To prepare data, we crop about 2 million images to $256 \times 256$ regions and downsample them to $128 \times 128$. After tokenization, we get $32 \times 32$ and $16 \times 16$ sequence pairs for different resolution. The pattern of finetuning sequence is "`[ROI1]` text tokens `[BASE]` `[BOI1]` $16 \times 16$ image tokens `[EOI1]` `[ROI2]` `[BASE]` `[BOI2]` $32 \times 32$ image tokens `[EOI2]`", longer than the max position embedding index 1087. As a solution, we recount the position index from 0 at `[ROI2]`.[7]

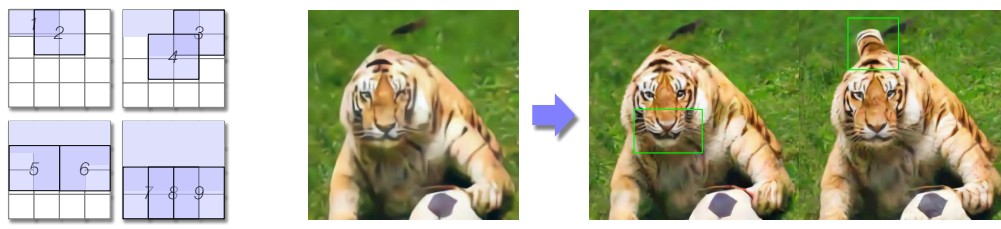

(a) Center–continuous sliding window      (b) Different super–resolution results for "a tiger is playing football".

Figure 5: (a) A $64 \times 64$-token image are generated patch-by-patch in the numerical order. The overlapping positions will *not* be overwritten. The key idea is to make the tokens in the 2nd and 4th regions – usually regions of faces or other important parts – generated when attending to the whole region. (b) The finetuned super-resolution model does not barely transform the textures, but generates new local structures, e.g. the open mouth or tail in the example.

---

[6]An evidence to support the belief is that if we append "close-up view" at the end of the text, the model will generate details of a part of the object.

[7]One might worry about that the reuse of position indices could cause confusions, but in practice, the model can distinguish the two images well, probably based on whether they can attend to a `[ROI2]` in front.

### 3.2 Image Captioning and Self-reranking

To finetune CogView for image captioning is straightforward: exchanging the order of text and image tokens in the input sequences. Since the model has already learnt the corresponding relationships between text and images, reversing the generation is not hard. We did not evaluate the performance due to that (1) there is no authoritative Chinese image captioning benchmark (2) image captioning is not the focus of this work. The main purpose of finetuning such a model is for self-reranking.

We propose the *Caption Loss* (CapLoss) to evaluate the correspondence between images and text. More specifically, $\text{CapLoss}(x,t) = \frac{1}{|t|}\sum_{i=0}^{|t|} -\log p(t_i|x, t_{0:i-1})$, where $t$ is a sequence of text tokens and $x$ is the image. $\text{CapLoss}(x,t)$ is the cross-entropy loss for the text tokens, and this method can be seen as an adaptation of inverse prompting [56] for text-to-image generation. Finally, images with the lowest CapLosses are chosen.

Compared to additionally training another constrastive self-supervised model, e.g. CLIP [38], for reranking, our method consumes less computational resource because we only need finetuning. The results in Figure 9 shows the images selected by our methods performs better in FID than those selected by CLIP. Figure 6 shows an example for reranking.

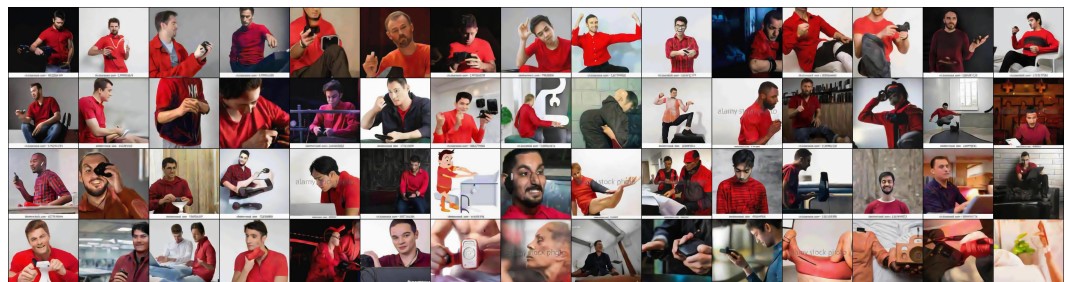

Figure 6: 60 generated images for "A man in red shirt is playing video games" (selected at random from COCO), displayed in the order of CapLoss. Most bad cases are ranked in last places. The diversity also eases the concern that CogView might be overfitting a similar image in the training set.

### 3.3 Style Learning

Although CogView is pretrained to cover diverse images as possible, the desire to generate images of a specific style or topic cannot be satisfied well. We finetune models on four styles: Chinese traditional drawing, oil painting, sketch, and cartoon. Images of these styles are automatically extracted from search engine pages including Google, Baidu and Bing, etc., with keyword as "An image of {style} style", where {style} is the name of style. We finetune the model for different styles separately, with 1,000 images each.

During finetuning, the corresponding text for the images are also "An image of {style} style". When generating, the text is "A {object} of {style} style", where {object} is the object to generate. In this way, CogView can transfer the knowledge of shape of the objects learned from pretraining to the style of finetuning. Figure 7 shows examples for the styles.

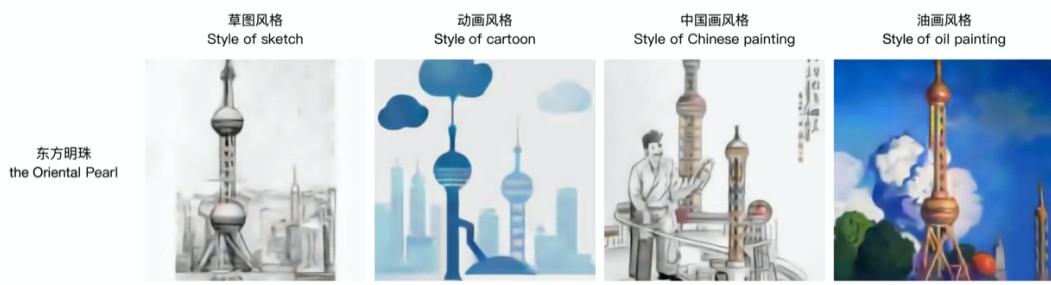

Figure 7: Generated images for "The Oriental Pearl" (a landmark of Shanghai) in different styles.

### 3.4 Industrial Fashion Design

When the generation targets at a single domain, the complexity of the textures are largely reduced. In these scenarios, we can (1) train a VQGAN [15] instead of VQVAE for the latent variable for more realistic textures, (2) decrease the number of parameters and increase the length of sequences for a higher resolution. Our *three-region sparse attention* (Appendix B) can speed up the generation of high-resolution images in this case.

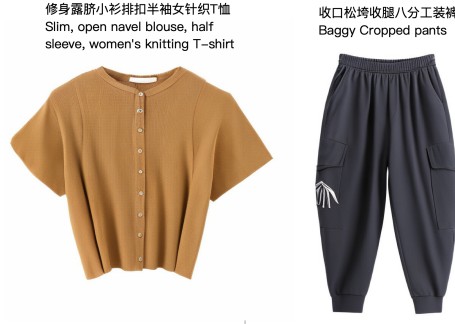

修身露脐小衫排扣半袖女针织T恤
Slim, open navel blouse, half sleeve, women's knitting T–shirt

收口松垮收腿八分工装裤
Baggy Cropped pants

We train a 3B-parameter model on about 10 million fashion-caption pairs, using $50 \times 50$ VQGAN image tokens and decodes them into $800 \times 800$ pixels. Figure 8 shows samples of CogView for fashion design, which has been successfully deployed to Alibaba Rhino fashion production.

Figure 8: Generated images for fashion design.

## 4 Experimental Results

### 4.1 Machine Evaluation

At present, the most authoritative machine evaluation metrics for general-domain text-to-image generation is the FID on MS COCO, which is not included in our training set. To compare with DALL-E, we follow the same setting, evaluating CogView on a subset of 30,000 captions sampled from the dataset, after applying a Gaussian filter with varying radius to both the ground-truth and generated images.[8] The captions are translated into Chinese for CogView by machine translation. To fairly compare with DALL-E, we do not use super-resolution. Besides, DALL-E generates 512 images for each caption and selects the best one by CLIP, which needs to generate about 15 billion tokens. To save computational resource, we select the best one from 60 generated images according to their CapLosses. The evaluation of CapLoss is on a subset of 5,000 images. We finally enhance the contrast of generated images by 1.5. Table 1 shows the metrics for CogView and other methods.

Table 1: Metrics for machine evaluation. Statistics about DALL-E and GANs are extracted from their figures. FID-$k$ means that all the images are blurred by a Gaussian Filter with radius $k$.

| Model | FID-0 | FID-1 | FID-2 | FID-4 | FID-8 | IS | CapLoss |
|-------|-------|-------|-------|-------|-------|------|---------|
| AttnGAN | 35.2 | 44.0 | 72.0 | 108.0 | 100.0 | 23.3 | 3.01 |
| DM-GAN | **26.5** | 39.0 | 73.0 | 119.0 | 112.3 | **32.2** | 2.87 |
| DF-GAN | **26.5** | 33.8 | 55.9 | 91.0 | 97.0 | 18.7 | 3.09 |
| DALL-E | 27.5 | 28.0 | 45.5 | 83.5 | 85.0 | 17.9 | — |
| CogView | 27.1 | **19.4** | **13.9** | **19.4** | **23.6** | 18.2 | **2.43** |

**Caption Loss as a Metric.** FID and IS are designed to measure the quality of unconditional generation from relatively simple distributions, usually single objects. However, text-to-image generation should be evaluated pair-by-pair. Table 1 shows that DM-GAN achieves the best unblurred FID and IS, but is ranked last in human preference (Figure 10(a)). Caption Loss is an absolute (instead of relative, like CLIP) score, so that it can be averaged across samples. It should be a better metrics for this task and is more consistent with the overall scores of our human evaluation in § 4.2.

**Comparing self-reranking with CLIP.** We evaluate the FID-0 and IS of CogView-generated images selected by CLIP and self-reranking on MS COCO. Figure 9 shows the curves with different number of candidates. Self-reranking gets better FID, and steadily refines FID as the number of candidates increases. CLIP performs better in increasing IS, but as discussed above, it is not a suitable metric for this task.

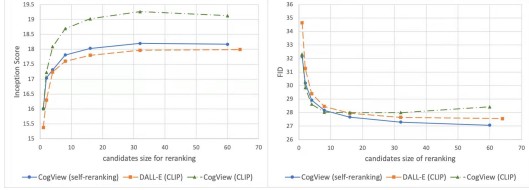

Figure 9: IS and FID-0 for CLIP and self-ranking.

---

[8]We use the same evaluation codes with DM-GAN and DALL-E, which is available at `https://github.com/MinfengZhu/DM-GAN`.

**Discussion about the differences in performance between CogView and DALL-E.** Since DALL-E is pretrained with more data and parameters than CogView, why CogView gets a better FID even without super-resolution? It is hard to know the accurate reason, because DALL-E is not open-source, but we guess that the reasons include: (1) CogView uses PB-relax and Sandwich-LN for a more stable optimization. (2) DALL-E uses many cartoon and rendered data, making the texture of generated images quite different from that of the photos in MS COCO. (3) Self-reranking selects images better in FID than CLIP. (4) CogView is trained longer (96B trained tokens in CogView vs. 56B trained tokens in DALL-E).

## 4.2 Human Evaluation

Human evaluation is much more persuasive than machine evaluation on text-to-image generation. Our human evaluation consists of 2,950 groups of comparison between images generated by AttnGAN, DM-GAN, DF-GAN, CogView, and recovered ground truth, i.e., the ground truth blurred by our image tokenizer. Details and example-based comparison between models are in Appendix E.

Results in Figure 10 show that CogView outperforms GAN-based baselines at a large margin. CogView is chosen as the best one with probability 37.02%, competitive with the performance of recovered ground truth (59.53%). Figure 10(b)(c) also indicates our super-resolution model consistently improves the quality of images, especially the clarity, which even outperforms the recovered ground truth.

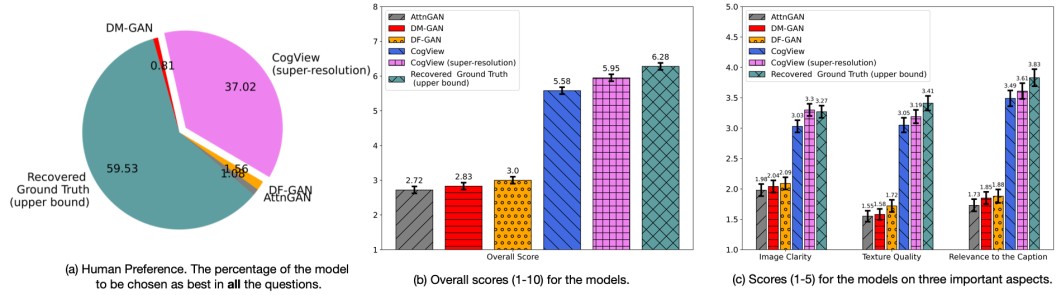

Figure 10: Human Evaluation results. The recovered ground truth is obtained by first encoding the ground truth image and then decoding it, which is theoretically the upper bound of CogView.

## 5 Conclusion and Discussion

**Limitations.** A disadvantage of CogView is the slow generation, which is common for auto-regressive model, because each image is generated token-by-token. The blurriness brought by VQVAE is also an important limitation. These problems will be solved in the future work.

**Ethics Concerns.** Similar to Deepfake, CogView is vulnerable to malicious use [49] because of its controllable and strong capacity to generate images. The possible methods to mitigate this issue are discussed in a survey [5]. Moreover, there are usually fairness problems in generative models about human [9]. In Appendix D, we analyze the situation about fairness in CogView and introduce a simple "word replacing" method to solve this problem.

We systematically investigate the framework of combining VQVAE and Transformers for text-to-image generation. CogView demonstrates promising results for scalable cross-modal generative pretraining, and also reveals and solves the precision problems probably originating from data heterogeneity. We also introduce methods to finetune CogView for diverse downstream tasks. We hope that CogView could advance both research and application of controllable image generation and cross-modal knowledge understanding, but need to prevent it from being used to create images for misinformation.

---

[9] https://thegradient.pub/pulse-lessons

## Acknowledgments and Disclosure of Funding

We would like to thank Zhao Xue, Zhengxiao Du, Hanxiao Qu, Hanyu Zhao, Sha Yuan, Yukuo Cen, Xiao Liu, An Yang, Yiming Ju for their help in data, machine maintaining or discussion. We would also thank Zhilin Yang for presenting this work at the conference of BAAI.

Funding in direct support of this work: a fund for GPUs donated by BAAI, a research fund from Alibaba Group, NSFC for Distinguished Young Scholar (61825602), NSFC (61836013).

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
