## A    Data Collection and Details about the Tokenizers

We collected about 30 million text-image pairs from multiple channels, and built a 2.5TB new dataset (after tokenization, the size becomes about 250GB). The dataset is an extension of project WudaoCorpora [52][10]. About 50% of the text is in English, including Conceptual Captions [44]. They are translated into Chinese by machine translation. In addition, we did not remove the watermarks and white edges in the dataset even though they affect the quality of generated images, because we think it will not influence the conclusions of our paper from the perspective of research.

The sources of data are basically classified into the following categories: (1) Professional image websites (both English and Chinese). The images in the websites are usually with captions. Data from this channel constitute the highest proportion. (2) Conceptual Captions [44] and ImageNet [11]. (3) News pictures online with their surrounding text. (4) A small part of item-caption pairs from Alibaba . (5) Image search engines. In order to cover as many common entities as possible, we made a query list consist of 1,200 queries. Every query was an entity name extracted from a large-scale knowledge graph. We choose seven major categories: food, regions, species, people names, scenic, products and artistic works. We extracted top-$k$ entities for each category based on their number of occurrences in the English Wikipedia, where $k$ is manually selected for each category. We collected the top-100 images returned by every major search engine website for each query.

We have already introduced tokenizers in section 2.2, and here are some details. The text tokenizer are directly based on the SentencePiece package at `https://github.com/google/sentencepiece`. The encoder in the image tokenizer is a 4-layer convolutional neural network (CNN) with 512 hidden units and ReLU activation each layer. The first three layers have a receptive field of 4 and stride of 2 to half the width and height of images, and the final layer is a $1 \times 1$ convolution to transform the number of channels to 256, which is the hidden size of embeddings in the dictionary. The decoder have the same architecture with the encoder except replacing convolution as deconvolution. The embeddings in the dictionary are initialized via Xavier uniform initialization [18].

## B    Sparse Attention

As shown in Figure 11, we design the *three-region sparse attention*, an implementation-friendly sparse attention for text-to-image generation. Each token attends to all text tokens, all *pivots* tokens and tokens in the blocks in an adjacent window before it.

The pivot tokens are image tokens selected at random, similar to big bird [53]. They are re-sampled every time we enter a new layer. We think they can provide global information about the image.

The blockwise window attention provides local information, which is the most important region. The forward computation of 1-D window attention can be efficiently implemented inplace by carefully padding and altering the strides of tensors, because the positions to be attended are already continuous in memory. However, we still need extra memory for backward computation if without customized CUDA kernels. We alleviate this problem by grouping adjacent tokens into blocks, in which all the tokens attend to the same tokens (before causally masking). More details are included in our released codes.

In our benchmarking on sequences of 4096 tokens, the three-region sparse attention (768 text and pivot tokens, 768 blockwise window tokens) is $2.5\times$ faster than vanilla attention, and saves $40\%$ GPU memory.

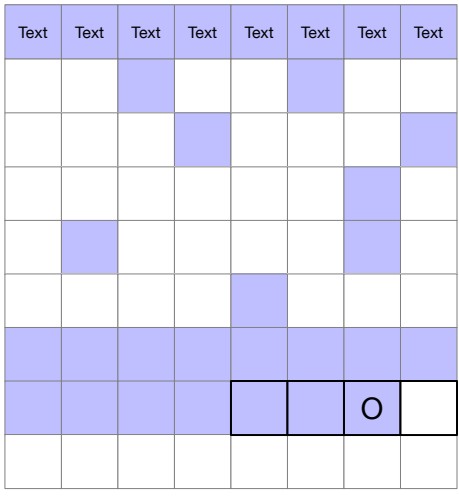

Figure 11: Illustration about our three-region sparse attention. The sequence is shown as a $H \times W$ image and some text tokens in front. Colored grids are all the tokens attended to by the token marked "O". In this case, each block consists of four consecutive tokens.

[10] `https://wudaoai.cn/data`

The whole training is $1.5\times$ faster than that with vanilla attention and saves $20\%$ GPU memory. With the same hyperparameters, data and random seeds, their loss curves are nearly identical, which means the sparse attention will not influence the convergence.

However, we did not use three-region sparse attention during training the 4-billion-parameter CogView, due to the concern that it was probably not compatible with finetuning for super-resolution in section 3.1. But it successfully accelerated the training of CogView-fashion without side effects.

## C  Attention Analysis

To explore the attention mechanism of CogView, we visualize the attention distribution during inference by plotting heat maps and marking the most attended tokens. We discover that our model's attention heads exhibit strong ability on capturing both position and semantic information, and attention distribution varies among different layers. The analysis about the scale of attention scores is in section C.4.

### C.1  Positional Bias

The attention distribution is highly related to images' position structures. There are a lot of heads heavily attending to fixed positional offsets, especially multiple of 32 (which is the number of tokens a row contains) (Figure 12 (a)). Some heads are specialized to attending to the first few rows in the image (Figure12 (b)) . Some heads' heat maps show checkers pattern (Figure 12 (c)), indicating tokens at the boundary attends differently from that at the center. Deeper layers also show some broad structural bias. For example, some heads attend heavily on tokens at top/lower half or the center of images (Figure 12 (d)(e)).

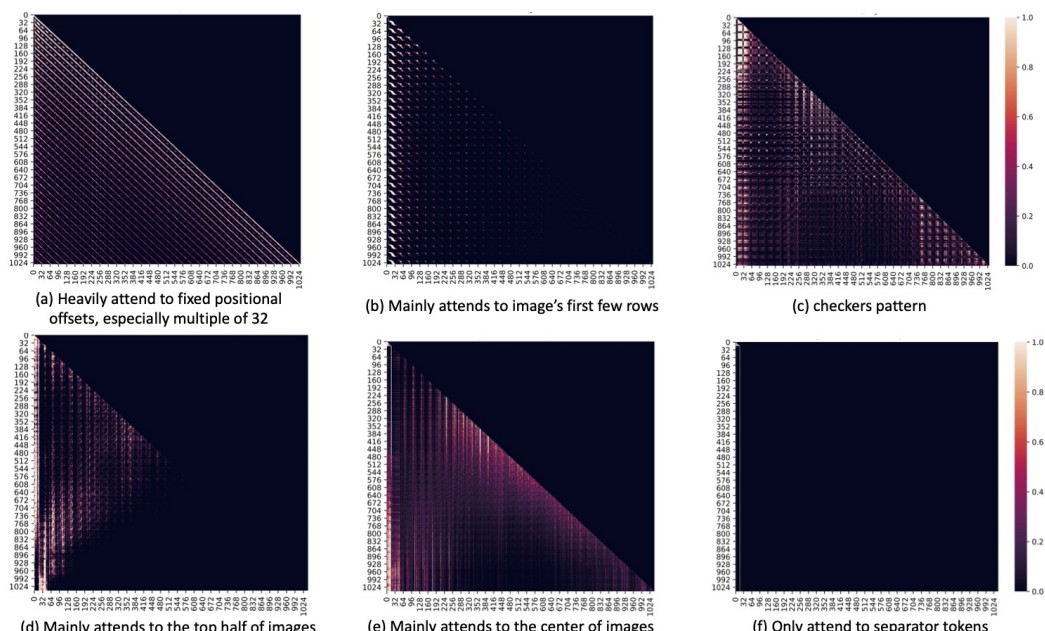

Figure 12: (a)(b)(c) Our model's attention is highly related to images' positional structures. (d)(e) Our model's attention show some broad structural bias. (f) Some heads only attend to a few tokens such as separator token.

### C.2  Semantic Segmentation

The attention in CogView also shows that it also performs implicit semantic segmentation. Some heads highlight major items mentioned in the text. We use "There is an apple on the table, and there is a vase beside it, with purple flowers in it." as input of our experiment. In Figure 13 we marked

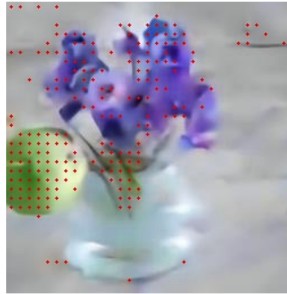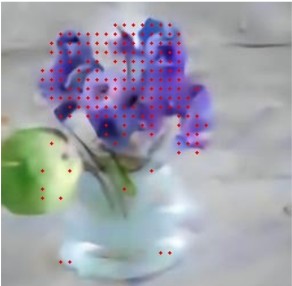

Figure 13: Our model's attention heads successfully captured items like apple and purple flowers. Pixels corresponding to the most highly attended tokens are marked with red dots.

pixels corresponding to the most highly attended tokens with red dots, and find that attention heads successfully captured items like apple and purple flowers.

## C.3 Attention Varies with Depth

Attention patterns varies among different layers. Earlier layers focus mostly on positional information, while later ones focus more on the content. Interestingly, we observe that attention become sparse in the last few layers (after layer 42), with a lot of heads only attend to a few tokens such as separator tokens (Figure 12 (f)). One possible explanation is that those last layers tend to concentrate on current token to determine the output token, and attention to separator tokens may be used as a no-op for attention heads which does not substantially change model's output, similar to the analysis in BERT [10]. As the result, the last layers' heads disregard most tokens and make the attention layers degenerate into feed-forward layers.

## C.4 Value Scales of Attention

As a supplement to section 2.4, we visualize the value scales of attention in the 38-th layer, which has the largest scale of attention scores $Q^T K/\sqrt{d}$ in CogView. The scales varies dramatically in different heads, but the variance in each single head is small (that is why the attention does not degenerate, even though the scores are large). We think the cause is that the model wants different *sensitiveness* in different heads, so that it learns to multiply different constants to get $Q$ and $K$. As a side effect, the values may have a large bias. The PB-relax for attention is to remove the bias during computation.

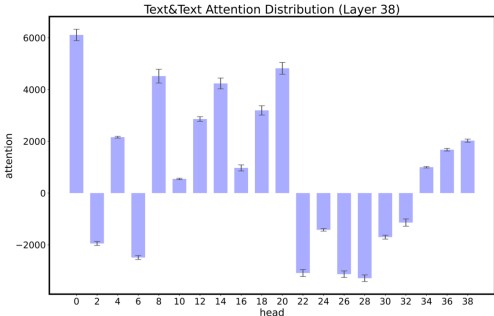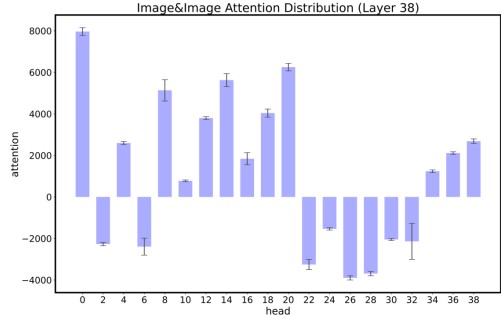

Figure 14: Illustration of scales of attention scores in the 38-th layer. Only half are heads are shown for display reasons. The error bar is from the minimum to the maximum of scores. The values of text-to-text attention scores are smaller, indicating the scales are related to the data.



Figure 15: The distribution of different genders, races and ages of the generation of "a face, photo".

## D  Fairness in CogView: Situation and Solution

**Evaluation of the situation of fairness in CogView.** We examine the bias in the proportion of different races and genders. Firstly, if given the detailed description in the text, e.g. a black man or an Asian woman, CogView can generate correctly for almost all samples. We also measure the proportion of the generated samples without specific description by the text "a face, photo". The figure of proportions in different races and genders are in Figure 15. The (unconditional) generated faces are relatively balanced in races and ages, but with more men than women due to the data distribution.

CogView is also beset by the bias in gender due to the stereotypes if not specifying the gender. However if we specify the gender, almost all the gender and occupation are correct. We tested the examples introduced in [6], and generated images for the text {male, female} × {"science", mathematics", "arts", "literature"}. Results are showed in this outer link to reduce the size of our paper.

**Word Replacing Solution.** Different from the previous unconditional generative models, we have a very simple and effective solution for racial and gender fairness.

We can directly add some adjective words sampled from "white", "black", "Asian", ..., and "male", "female" (if not specified) in the front of the words for human, like "people" or "person", in the text. The sampling is according to the real proportion in the whole population. We can train an additional NER model to find the words about human.

Since CogView will predict correctly according to the results above, if given description, this method will greatly help solve the fairness problem in generative models.

## E  Details about Human Evaluation

To evaluate the performance, we conduct a human evaluation to make comparisons between various methods, similar to previous works [27, 39]. In our designed evaluation, 50 images and their captions are randomly selected from the MS COCO dataset. For each image, we use the caption to generate images based on multiple models including AttnGAN, DM-GAN, DF-GAN and CogView. We do not generate images with DALL-E as their model has not been released yet. For each caption, evaluators are asked to give scores to 4 generated images and the recovered ground truth image respectively. The recovered ground truth image refers to the image obtained by first encoding the ground truth image (the original image in the MS COCO dataset after cropped into the target size) and then decoding it.

For each image, evaluators first need to give 3 scores ($1 \sim 5$) to evaluate the image quality from three aspects: the image clarity, the texture quality and the relevance to the caption. Then, evaluators will give an overall score ($1 \sim 10$) to the image. After all 5 images with the same caption are evaluated, evaluators are required to select the best image additionally.

72 anonymous evaluators are invited in the evaluation. To ensure the validity of the evaluation results, we only collect answers from evaluators who complete all questions and over 80% of the selected best images are accord with the one with the highest overall quality score. Finally, 59 evaluators are kept. Each evaluator is awarded with 150 yuan for the evaluation. There is no time limit for the answer.

To further evaluate the effectiveness of super-resolution, we also introduced a simple A-B test in the human evaluation. Evaluators and captions are randomly divided into two groups $E_a, E_b$ and $C_a, C_b$

respectively. For evaluators in $E_a$, the CogView images with captions from $C_a$ are generated without super-resolution while those from $C_b$ are generated with super-resolution. The evaluators in $E_b$ do the reverse. Finally, we collected equal number of evaluation results for CogView images with and without super-resolution.

The average scores and their standard deviation are plotted in Figure 10. Several examples of captions and images used in the human evaluation are listed in Figure 16. The evaluation website snapshots are displayed in Figure 17.

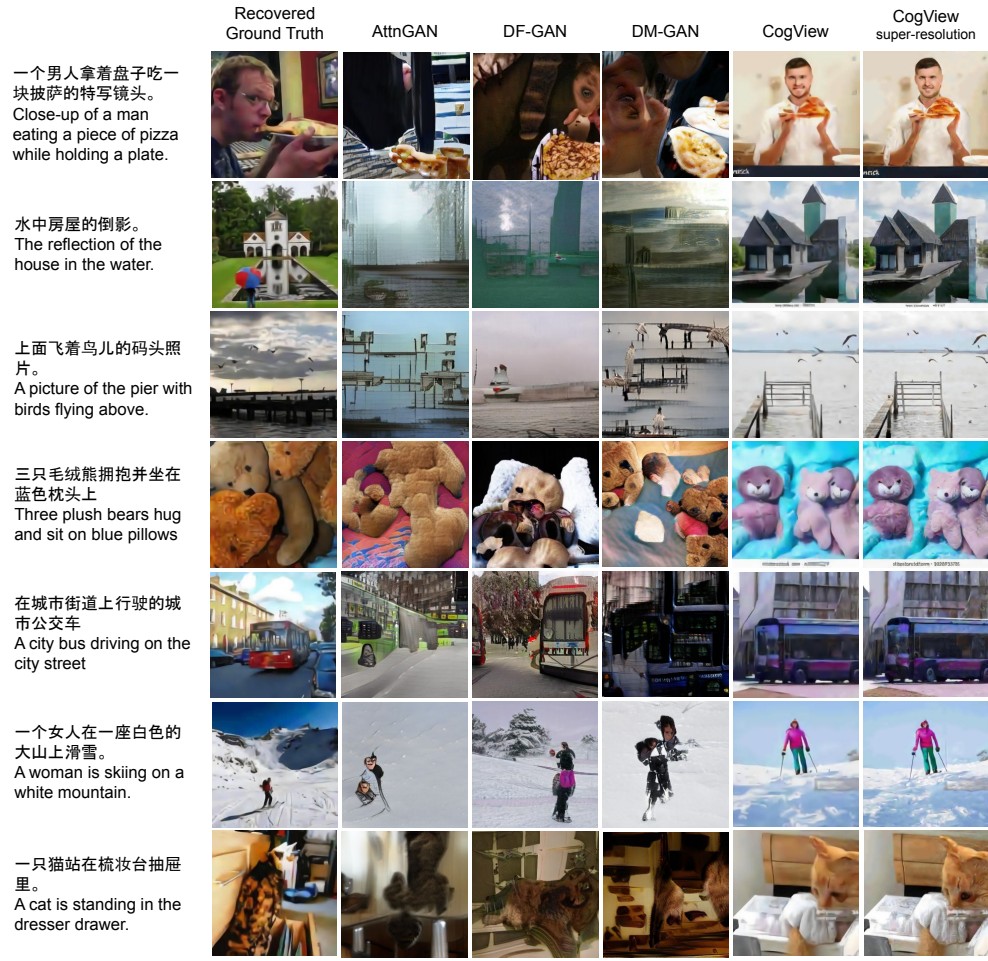

Figure 16: Human evaluation examples. The captions for evaluation are selected at random from MS COCO.

# F    Show Cases for captions from MS COCO

In Figure 18, we provide further examples of CogView on MS COCO.

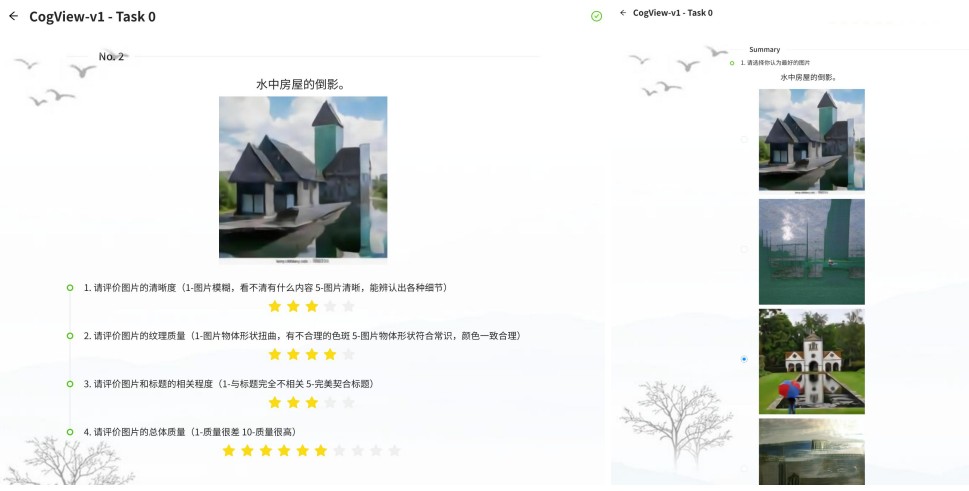

Figure 17: Snapshots of the human evaluation website. The left side is the scoring page for images and the right side is the best-selection page for all images with the same caption.

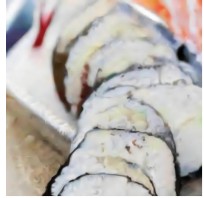
A plate of faux sushi sits on a restaurant table.

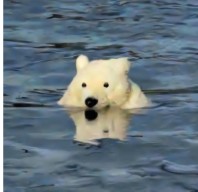
A young polar bear swims through icy waters.

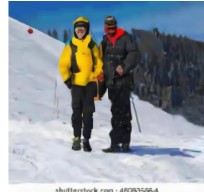
Two skiers pose beside each other on a slope.

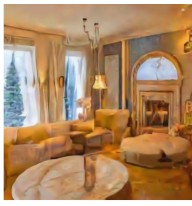
A very big and nice looking living room.

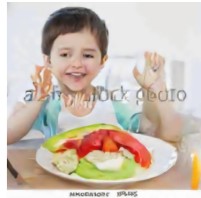
A boy is getting ready to eat some food from a plate.

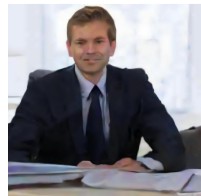
A man in a business suit sitting in front of paperwork.

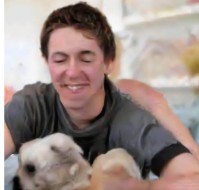
A man petting a small dog and smiling for a camera.

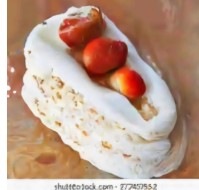
A piece of white chocolate and strawberry cake.

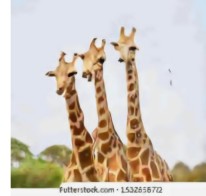
Three giraffes that are standing up near each other.

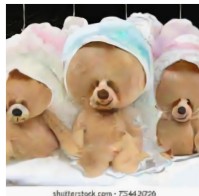
a couple of teddy bears that have cloths on

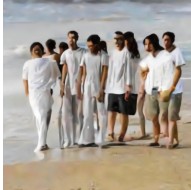
There are a lot of people that are at the beach.

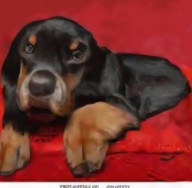
A brown and black dog laying on a red blanket.

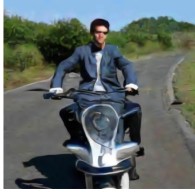
A man riding on the back of a motorcycle down a road.

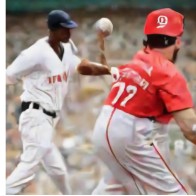
A man is up to bat in a professional baseball game.

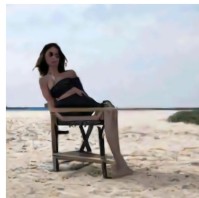
A woman sitting in a chair at the beach.

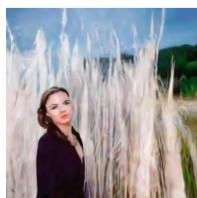
A woman in a black and purple dress poses in front of some tall grass.

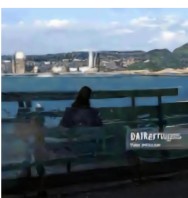
A woman is on a bench overlooking the city.

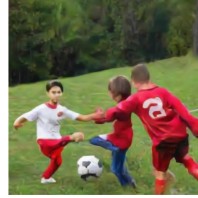
A couple of young boys playing a game of soccer.

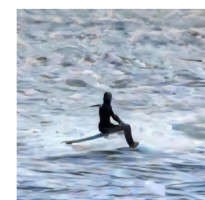
a man that is on a surfboard in some water.

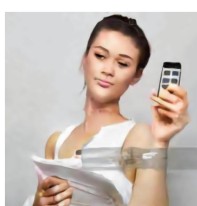
A women in a white blouse is holding a remote in her hands.

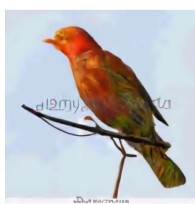
A bird perched on top of a leafless tree under a blue sky.

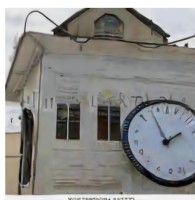
a clock hanging outside of a house in a nice neighborhood.

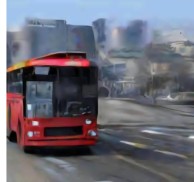
A red bus is driving on the road.

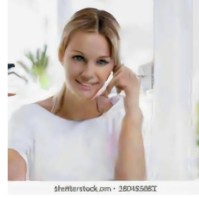
A beautiful young blond woman talking on a phone.

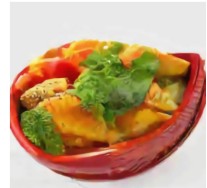
A red bowl filled with food and leafy greens.

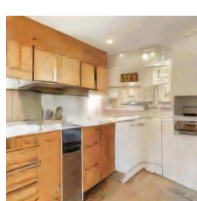
A small galley kitchen with wooden cabinets and white appliances.

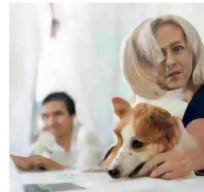
A woman sitting on a computer desk while holding her adorable dog.

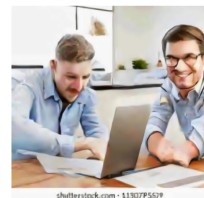
Two men are sitting at tables using laptop computers.

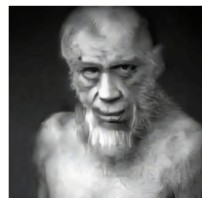
an image of a man dressed up like an ape

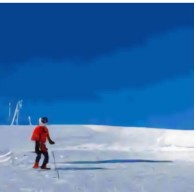
A man on a snowboard snowboarding on a mountain slope.

Figure 18: More generated images for COCO captions (after super-resolution).