# OpenReview forum: "CogView: Mastering Text-to-Image Generation via Transformers"
_NeurIPS.cc/2021/Conference — NeurIPS 2021 Poster_

### Official Review · Reviewer_TthA · 2021-07-12

**Rating:** 6
**Confidence:** 4

**Summary:**

- CogView is an text-to-image model similar to DALL-E, except it uses a traditional VQ-VAE instead of dVAE.
- A Chinese (image, text) dataset is collected for supervised pre-training. (30M pairs)
- A couple of techniques are described to stabilize FP16 training: PB-Relax (identity that is more stable for computing attention), Sandwich-LN (pre + post LayerNorm) .
- They fine-tune the model for style-learning, super-resolution, and image captioning, and fashion design, and show samples but do not provide quantitative results.
- CogView is compared (via FID/IS) to DALL-E on MS-COCO by translating captions to Chinese first.
- Human evaluation is used to compare DM-GAN, DF-GAN, AttnGan, and Ground-truth. 37% of the time, CogView is listed as the best.
- CaptionScore is proposed as an image-text metric, derived from the likelihood of text tokens assigned from CogView. This is also used to select the best txt2img candidate amon 60, unlike DALL-E which relies on a separate CLIP model.


**Limitations And Societal Impact:**

No, there is no discussion.

**Main Review:**

(Please read summary) While there are some interesting empirical findings, the contribution over DALL-E is limited. The primary difference is dataset collection, which focused primarily on photographs. There is no reason provided for collecting the dataset in Chinese, while much of the dataset is originally in English, but translated. The details of the collection are insufficient for reproducibility and it does not seem like this is a dataset paper either. The main reason given for better FID/IS scores on MS-COCO is that pre-training was conducted mainly on photos, which is more similar to MS-COCO, so the result is not surprising.

One missing baseline from the human evaluation is the retrieval one, e.g. 'closest image in training set' from a text2image retrieval system. It is unclear how much of the generated images is memorized/regurgitated.

I do like the idea of CaptionScore to rank generated images instead of a relying on another model.

Update: Increased score in light of discussion of potentially useful tricks to stabilize training contributed by this work, which may be useful to the community. The appendix has an expansive analysis on attention in the model that motivates PB-relax. Not having to rely on another model like CLIP is also nice.

**Time Spent Reviewing:**

3 hours

---

> ### Author Response · Authors · 2021-08-08
> **Authors' response**
>
> Thank you for your valuable comments, we conclude your concerns as following points and will explain them point by point.
>
> 1. "Lack of quantitative results for style-learning, super-resolution, and image captioning/reranking, and fashion design". (Summary 4)
>
>    This is not accurate. We **indeed carefully demonstrate quantitative results** (large-scale human evaluation scores) for super-resolution in Figure 8 (a)(b)(c) and analyze them in Line 294-296. There is no comparison with CogView-super-resolution and DALL-E on FID, because the FID scores with different resolution cannot be directly compared. Human evaluation is already the most convincing metric for this generation task.
>
> For image ranking, which is only the final step of generation, the quantitative results are included in the results in Table 1. Here we also give a more detailed quantative results of ranking for N candidates (without super-resolution) and its comparison with DALL-E:
>
> The FID-0 of reranking from 1, 2, 4, 8, 16, 32, 60 images for CogView(self-reranking post-selection), DALL-E(CLIIP) and CogView(CLIP): https://i.imgur.com/UX9H022.jpg
>
> The IS of reranking from 1, 2, 4, 8, 16, 32, 60 images for CogView(self-reranking post-selection), DALL-E(CLIIP) and CogView(CLIP): https://i.imgur.com/4ajxZ0f.jpg
>
> It seems that the CLIP reranking are better on IS, and self-reranking are better on FID. FID is more consistent with human evaluation in our experiments.
>
>    For fashion design and style-learning, they are novel and practical applications, lack of benchmark and baselines, but the fashion design has already got into industrial use, which also shows its efficacy.
>
> 2. "The improvement is totally owed to dataset collection (mainly on photos),  so the result is not surprising."
>
>    This is a misunderstanding. The improvements of CogView over DALL-E are two folds:
>
>    1. Finetuning for super-resolution. **This is totally new and proposed by our paper, and it largely increases the generation quality over zero-shot generation**. See Figure 5 for qualitative results and Figure 8 (a)(b)(c) for quantative results.
>
>    2. Even without  super-resolution finetuning, our pretrained model outperforms DALL-E on FID of MS COCO. The results are in Table 1. The reasons could be mixed, and we will never know them exactly because DALL-E is close-source. However,  we think the main reason could be that our Sandwich-LN and PB-relax could be a better stabilization method to improve training, because **CogView only uses 1/5 data and 1/3 parameters of DALL-E**. The other possible reasons are the data distribution and training duration.
>
>    3. We are sorry that there is a typo in Line 277 to **miss "Besides the different normalization achitecture in the Transformer Layer,"** before "We guess the main reason is that our data ... mainly photos."  What we actually want to express is that, even if the data of CogView only 1/5 of that of DALL-E, as researchers, we cannot directly owe the success totally to our new techniques instead of data, because the data distribution could be also important. As we find CogView can also generate good cartoons, pixel-style images, etc. with proper prompt after submission, the data are actually not mainly photos and we will rewrite this paragraph later.
>
>       If the misunderstanding caused by this typo is the only reason for you to make this claim, we will greatly appreciate it if you can increase your rating to 6 or above after we clarifying it here.
>
>
>
> 3. "Why did you train a Chinese model, instead of English?"
>
>    Chinese is the mother language of the authors and the language with most speakers in the world, so it is very natural for us to train a Chinese model for more people to use when there are already (slightly) more than half resources are original in Chinese. If you are worrying about the comparison, we translate the titles from MS COCO into Chinese and this should only decrease the performance of CogView instead of DALL-E. If you are worrying about the usage from other countries, we have already added an auto-translation button on our demo website and serves many users from China, America, Russia, Japan, etc.
>
> 4. "In human evaluation, a baseline of the most similar image in training data is lacked. The model might simply remember some images."
>
>    Firstly, in Line 249 we already discussed the concerns about "remembering the training data". As an evidence, the generated samples (e.g. Figure 6) are very diverse for all the texts, it is impossible to have so many similar samples to remember in the training set. This concern is usually for GAN-based model, which is suffered from mode collapse, while is not serious for auto-regressive model. At each token, different sampling will largely change the generation afterwards. How can such a model copy an existing image?
>
>    Secondly, to add such a baseline to human evaluation seems not helpful to solve the concern. People will give a very good realistic score to it because it is a real image. And its relevance score can be used to assess the text-to-image retrieval model, but it is not very relevant to the generation task.
>
>    Instead, we will plan to train another constrastive learning model to find similar images in the dataset to compare with the generated samples to further ease your concerns in the camera-ready version if accepted.
>
> 5. "There is no contribution over DALL-E."
>
>    In the introduction section, we list 4 main contributions over DALL-E, which are also the main contents of this paper. Here, we will stress **why the 4 contributions are important**.
>
>    ​	Contribution 1: New transformer normalization backbone Sandwich-LN and PB-Relax. Although seeming to be an add-on at the first glance,  to stabilize of FP16 pertraining is the core and most challenging part in training text-to-image generation models, which is both argreed by CogView and  DALL-E (section 2.4 paragraph 1). Firstly, training large models usually relies on deepspeed ZERO (Line 157), which only supports FP16. Secondly, Sandwich-LN + PB-Relax is much more simple and effective than the specific mix-precision system of DALL-E (not very detailed and close-source), so our solution is the only method to train such a model currently.  Moreover, as discussed in the paper, the new normalization backbone additional stabilizes 10B NLP models and 64-layer toy model, meaning that its potential is beyond this task.
>
>    ​	Contribution 2: First to propose finetuning strategies for text-to-image generation. DALL-E leaves finetuning for future work (section 3.1 paragraph 5), while CogView proposes a series finetuning methods for different finetuning tasks. Among them, **super-resolution greatly increase the quality of text-to-image generation with very few additional computational resource.** And caption model can be used for post-selection without training an expensive CLIP. The finetuning only need 1 DGX machine for 1 day, but can benefit various task. This is also a very important contribution.
>
>    ​	Contribution 3: New evaluation metrics for text-to-image generation. We find the drawback of previous metrics according to human evaluation, and propose a new metrics CapS, which is based on CogView Caption model and corresponds the human evaluation well. This will prevent the development of this topic from being hindered by improper metrics in the future.
>
>    ​	Contribution 4. Better performance and open-sourcing. Although both   target at text-to-image generation, CogView (base) achieves better scores, and CogView-finetuned (super-resolution) performs even much better. DALL-E is close-source, which makes us hard to perform ablation study to find which part is the main factor for Cogview to outperform it. So we really need an open-source pretrained text-to-image model to benefit the development of this topic -- which is CogView. This is a big contribution.
>
>    ​	Other contributions: (1) The three-region sparse attention with 2.5x faster and 0.6x memory without hurting performance in Appendix B. (2) A thorough investigation and visualization of the attention in the text-to-image generation tasks in Appendix C. (3) A useful summary and comparison of VQVAE tokenizer in the section 2.2. (4) the usage of VQGAN in text-to-image generation for the first time in the section 3.4. These contributions are not the main points of this paper but are also very helpful to the researchers in this area.
>
> We find the opinion of the reviewer might be mainly affected by some misunderstandings. If our answers above can successfully clarify them, could you increase your rating to 6?

---

> > ### Comment · Reviewer_TthA · 2021-08-17
> > **Response to response**
> >
> > > "The improvement is totally owed to dataset collection (mainly on photos), so the result is not surprising."
> >
> > Just wanted to clarify that this is not a direct quote from my review. However, this is a paraphrase from the paper. Throughout the author's response, quotes ("") are used for things that are *not* my quotes. Please avoid this.
> >
> > > "Chinese is the mother language of the authors and the language with most speakers in the world"
> >
> > This is a practical reason for the authors, but scientifically it is unjustified when the target evaluation data are in English, e.g. MS-COCO. It is unclear what the effect of the translation model is on performance of the text-to-image system. It is also unclear how much of gains in performance are due to dataset difference vs stabilization tricks.
> >
> > >  " it is impossible to have so many similar samples to remember in the training set"; "How can such a model copy an existing image?"
> >
> > It is very easy for a large Transformer language model to memorize full sequences and has been shown in numerous papers to be the case. In these types of models it is always interesting to know how much performance is due to memorization vs generalization.
> >
> > > ", the new normalization backbone additional stabilizes 10B NLP models and 64-layer toy model, meaning that its potential is beyond this task."
> >
> > No evidence for this was provided in the paper. A general comment: many statements like this in the paper lack empirical justification and are simply stated.

---

### Official Review · Reviewer_barm · 2021-07-15

**Rating:** 6
**Confidence:** 4

**Summary:**

This paper proposes a generative, two-stage text-image model that combines a VQVAE (or VQGAN) and a text-conditional autoregressive transformer model for text-to-image synthesis. Here, the VQ-model is trained on image data only (first stage), while in the second stage the transformer is trained to model the distribution of learned discrete image representations (="image tokens") of the (now fixed) first-stage model, given a corresponding textual description. The generated image tokens can then be decoded into the image space via the pre-trained decoder. Additionally, the fine-tuning of the model for specific tasks such as style learning, super-resolution, and image-to-text is explored. The work also provides insights into stabilizing the training of large transformers for text-image synthesis.

**Limitations And Societal Impact:**

The conclusion mentions that the proposed method could potentially be used to spread misinformation, which is a valid concern. Possible biases of the model or the collected dataset are not discussed. Limitations are rarely discussed, see also my above concerns about the writing.

**Main Review:**

__Strengths:__
The paper provides (very) good qualitative results. The results depicted in Fig. 1 and Fig. 15 are impressive.
Further, interesting insights on training of the VQVAE are presented. For example, apparently a fixed codebook is not much different from a learned one and codebook collapse does not seem to be an issue. (l.119 and l.128-131).

Moreover, the paper gives useful insights on how to stabilize the transformer training (OBS and Sandwich Layer Norm). Additionally, it is interesting that weighing the text loss and the image-token loss equally seems to perform best.
Excluding the dataset-collection, the proposed approach (or rather the model's architecture)  is reproducible from the description provided in the paper.

__Weaknesses:__

- Empirical justifications of the above statements about VQVAE and Transformer trainining is missing. The proposed methods are simply described as "working well". The same holds for the statements about the weighing of the text loss and the "center-continuous sliding window". The latter needs to be compared to ordering that do not allow for inconsistencies between tokens, i.e token 4 & 7 and 5 & 6 in Fig. 5a (counting from 1 to 16, raster-scan order).
- Evaluation is only done for a fixed re-ranking factor of 60. How does the model perform when re-ranking with less images (similar to Fig. 9 in [1])? How does the model perform when re-ranking is performed with CLIP?
- Very few qualitative examples given for the finetuning, especially "Style Learning".
- The writing should be improved. I would recommend the authors tone down some of their claims (e.g. l. 50: "Our approach is neat and natural", or in the title "mastering") and additionally discuss limitations (e.g. blurriness, sampling speed) of their approach (more) detailed.

__Mixed Comments & Questions:__
- What is the inference time (with and without the superres. model)? How does this compare to other approaches?
- l. 287-288: What does "Samples from CogView itself might be overscored, but it is all right to evaluate other models in the future" mean?
- l. 277-l.281 speculate about the reasons why CogView performs better than DALL-E on blurred MS-COCO. Comparing the first stage models (i.e. the discrete autoencoders) via reconstruction metrics could serve as an objective empirical justification for these claims.
- Section 3.1: Only a single example for style learning is given. Furthermore, only a single realization for each text prompt is depicted. How large is the variance given a single prompt? These question could be easily answered by plotting multiple realizations for each prompt.
- Section 3.3: Could the model be finetuned by reversing the order (i.e. predict text tokens from image tokens from right to left) in the autoregressive NLL training? This would correspond to the proposed "swapping" of text and image tokens, with additional swapping of the (pretrained) positional embeddings.
- Section 3.4: The structure of the paper suggests that this section still belongs to the "Finetuning" chapter, but, at least in my understanding, introduces (i) a new dataset and (ii) another first-stage model (VQGAN), and thus trains a new model from scratch. This should be clarified. To stay with this theme: Why is the VQGAN not used for the "big" data set? Since the resolution is very high with 50x50 tokens, hardly any compression artifacts should be visible according to [2]. Are there other ways to fix the bluriness of the generated samples? How are the massively increased compute requirements (i.e. caused by switching from 32x32 tokens to 50x50 tokens) handled?
- Why the name "CogView"? I could not find an explanation in the paper.
- How does the model perform in a "zero-shot" setting on other classic text-to-image datasets such as CUB [3]? Can the model be fine-tuned on these datasets?
- What is the stopping criterion when training the transformer? Minimal NLL on test data? Something else?
- l.106: which "chinese corpus"?
- Which model is used in Fig. 11?

_Minor_:

- l. 54: missing word, maybe "our" or "the"?
- l. 94: $z_i$s -> $z_i$
- l. 149: consine -> cosine
- l. 210: is -> are
- l. 523: alleviates -> alleviate

__Summary/Justification of Rating:__

In summary, I have somewhat mixed feelings about the paper. On the one hand, it (i) shows that two-stage training with VQVAE/VQGAN and Transformers can produce highly complex scenes from text conditioning, and provides convincing results (also demonstrated through a user study), and (ii) provides interesting insights into training the VQVAE and Transformer models (see above). On the other hand, many of the proposed modeling decisions feel somewhat "ad-hoc" without being justified by ablation experiments (see _weaknesses_ and _comments_ above). There are also no results supporting the claims about non-present codebook collapse (l.119) or "fixed/learnable embeddings" (l.128-131), which I think are of great interest to the community. In addition, the writing of the paper needs improvement and the limitations of the work need to be discussed. I am currently rating this paper as "just above threshold" but am willing to increase my score if the authors provide satisfactory answers.

__References:__
 - [1]: Ramesh, Aditya, et al. "Zero-shot text-to-image generation." arXiv preprint arXiv:2102.12092 (2021).
 - [2]: Esser, Patrick, Robin Rombach, and Bjorn Ommer. "Taming transformers for high-resolution image synthesis." Proceedings of the IEEE/CVF Conference on Computer Vision and Pattern Recognition. 2021.
 - [3]: Welinder, Peter, et al. "Caltech-UCSD birds 200." (2010).

**Time Spent Reviewing:**

10.5

---

> ### Author Response · Authors · 2021-08-08
> **Authors' response**
>
> Thank you very much for your careful and valuable comments, we will explain your concerns point by point.  Some results you wanted (e.g. Weaknness 2, 3) were originally in the paper and were removed from our draft due to limit of space. Here we show the results and will add them back to the camera-ready version if accepted.
>
> Weakness1：Empirical justifications of proposed "VQVAE (codebook training strategies)", "Transformer training" and center-continuous sliding window is missing.
>
> * For "Transformer training", I'm afraid that your comment is not very accurate. We have already showed the training dynamics of 5 different settings for ablation study to prove the effectiveness of our approaches in Figure 3(b) and Line 195-199 in the toy experiments. This is a slightly small setting with 64 layers and 1024 hidden units and deliberately inappropriate hyperparameters.
>
>   For CogView setting(48 layers, 2560 hidden units), the baseline (ordinary Pre-LN Transformer)  cannot be trained at all (same as the toy experiments), which will quickly go into unskipable overflow (Line 158-160).
>
>   As for the solution of DALL-E, a specific mix-precision system, is not open-source. Moreover,  we find it difficult to replicate the method merely based on the description in the DALL-E's paper. Thirdly, this work is actually concurrent with DALL-E. When the paper of DALL-E released, the pretraining of CogView had already begun for a long time. Since each pretraining comsumes a lot of money (about $300,000), it is not possible to change our stabilization method to re-train.
>
> * For "VQVAE (codebook training strategies)", we are re-running the experiments and will reply the loss curves in the session as soon as possible.
>
> * For "center-continuous sliding window", we are recruiting people for human evaluation. If we cannot finish them until the end of rebuttal, we will add them into the camera-ready version if accepted.
>
> Weakness 2: "Evaluation is only done for a fixed re-ranking factor of 60. How does the model perform when re-ranking with less images (similar to Fig. 9 in [1])? How does the model perform when re-ranking is performed with CLIP?"
>
> The FID-0 of reranking from 1, 2, 4, 8, 16, 32, 60 images for CogView(self-reranking post-selection), DALL-E(CLIIP) and CogView(CLIP):  https://i.imgur.com/UX9H022.jpg
>
> The IS of reranking from 1, 2, 4, 8, 16, 32, 60 images for CogView(self-reranking post-selection), DALL-E(CLIIP) and CogView(CLIP):  https://i.imgur.com/4ajxZ0f.jpg
>
> It seems that the CLIP reranking are better on IS, and self-reranking are better on FID. FID is more consistent with human evaluation in our experiments.
>
> Weakness 3: "Very few qualitative examples given for the finetuning, especially 'Style Learning'."
>
> More samples for sketch: https://i.imgur.com/QlF8ao1.jpg
>
> More samples for Chinese painting: https://i.imgur.com/tlS4Kes.jpg
>
> More samples for Catoon: https://i.imgur.com/TTT8TEr.jpg
>
> More samples for Oil painting: https://i.imgur.com/10ONJEl.jpg
>
> Weakness 4: "The writing should be improved. "
>
> Thank you very much for your advices. We will modify the claims and add discussion about blurriness and speed in the camera-ready version if accepted.
>
> ======================== For Questions ========================
>
> Question1: "What is the inference time (with and without the superres. model)? How does this compare to other approaches?"
>
> On a single A100 GPU, it needs 124s to generate a batch of 12 samples, and 221s for super-resolution (for 1 given image from the CogView zero-shot model).
>
> CogView is much slower than previous GAN-based model, which is a drawback of the auto-regressive models. The tokens cannot be generated in parallel, which makes  the inference much slower than training. However, CogView should be at least 3x faster than DALL-E, which have about 3x parameters of CogView.
>
> To accelerate the generation is left for future work.
>
> Question 2: "l. 287-288: What does "Samples from CogView itself might be overscored, but it is all right to evaluate other models in the future" mean?"
>
> The scoring model is finetuned from CogView, so it might learns similar things with CogView. So it is possible that scoring model will give a relative high score for CogView. But if two models are not relevant with CogView, this bias does not exist.
>
> Question 3: "l. 277-l.281 speculate about the reasons why CogView performs better than DALL-E on blurred MS-COCO. Comparing the first stage models (i.e. the discrete autoencoders) via reconstruction metrics could serve as an objective empirical justification for these claims."
>
> Firstly, we are sorry that there is a typo in Line 277 to miss "Besides the different normalization achitecture in the Transformer Layer," before "We guess the main reason is that our data ... mainly photos."  What we actually want to express is that, even if the data of CogView only 1/5 of that of DALL-E, as researchers, we cannot directly owe the success totally to our new structures (e.g. sandwich-LN) instead of data, because the data distribution is also important. As we find CogView can also generate cartoons, pixel-style images etc. with proper prompt after submission, we will rewrite this paragraph later.
>
> Secondly, we think the reviewer gives a good comparing method in the ideal situation, but unfortunately our tokenizer is trained only on part of the dataset, and it cannot represent the data distribution in the stage 2.
>
> Question 4: The same as Weakness 3. See the new samples.
>
> Question 5: " Could the model be finetuned by reversing the order (i.e. predict text tokens from image tokens from right to left) in the autoregressive NLL training? This would correspond to the proposed "swapping" of text and image tokens, with additional swapping of the (pretrained) positional embeddings."
>
> If we understand correctly, this method should be equivalent to our proposed method because they are both first-image-then-text generation, and their real positions in the input sequence does not affect the generation process. The positional embeddings should be easy to finetune because they are in every sequence.
>
> Question 6: "section 3.4: Why is the VQGAN not used for the "big" data set? Since the resolution is very high with 50x50 tokens, hardly any compression artifacts should be visible according to [2]. Are there other ways to fix the bluriness of the generated samples? How are the massively increased compute requirements (i.e. caused by switching from 32x32 tokens to 50x50 tokens) handled?"
>
> 1. VQGAN, or most currently GAN-based methods can only work on single domain. Unfortunately, we cannot find a reference to support this point, since the previous GAN-based works mainly reports the positive results on a single domain (e.g. Face, animal, church, bedroom etc.). We spent much efforts to train a VQGAN on the general domain but have not succeeded now. A possible reason is that the distribution of texture is very complex, it needs some global information to classify the context before generating the local texture, which is very hard for an ordinary small generator and classifier. In all, this is not a trivial problem and we are investigating the exact reasons, which should be in another paper.
>
> 2. We think that the blurrness roots in the lack of information at this level of resolution. (not strictly speaking,) VQGAN use *local mode collapse* to change the texture to a remembered texture in the dataset to increase the sharpness. The side effect of VQGAN is to generate strange textures, e.g. more eyes for animals on the bodies (e.g. the 2nd and 4th row, left column, Figure 4 in [2]).
>
>    In our opinion, the key to solve the blurrness is to do super-resolution finetuning to add new information  (or training a 64x64 model).
>
> 3. We have already account for the computation problem in Line 257-259 and Appendix. B, which introduces the 3-region sparse attention to speed up the training and reduce the memory consumption.
>
> Question 7: "Why the name "CogView"?"
>
> Cognitive + View. Just for the consistency with the names of previous works.
>
> Question 8: "How does the model perform in a "zero-shot" setting on other classic text-to-image datasets such as CUB [3]? Can the model be fine-tuned on these datasets?"
>
> We did not evaluate on CUB because we target at generation in general domain, which is more challenging and interesting. CogView can generate birds under zero-shot settings, and finetuning should also be useful, but the best solution should be VQGAN+Transformer, like what we did for fashion design. A good method in single domain usually performs not well in general domain, because of the difference in the complexity of distribution. That is why we did not spend much efforts on this dataset.
>
> Question 9: "What is the stopping criterion when training the transformer? Minimal NLL on test data? Something else?"
>
> Actually, we train the model until we ran out of budget (about $300,000). The training loss were still decreasing very slowly. Large model pretraining is hard to be trained to convergence, which is also the common case in NLP.
>
> Question 10: "l.106: which "chinese corpus"?"
>
> Large corpus crawled down from Zhihu and Baiduzhidao (two Chinese websites). We already open-sourced a superset of them, but cannot show the url due to double-blind review.
>
> Question  11: "Which model is used in Fig. 11?"
>
> The pretrained CogView model (without super-resolution).

---

> > ### Author Response · Authors · 2021-08-13
> > **Loss curves of training VQVAE under different settings**
> >
> > See https://i.imgur.com/zJlNZrL.jpg for loss curves. The architectures of VQVAE in these experiments are identical with that described in the paper. The training is on a subset of our dataset with about 1M images.
> >
> > As showed in the figure, the four settings converges to a very similar loss, which is discussed in the paper. We will add this figure to paper in the camera-ready version if accepted.
> >
> > As for the stage1 of DALL-E, the major difference (except dataset) with the gumbel-softmax setting is a KL loss with uniform distribution on the codes. In our experiments (before submission and recently), we find that this term only increases the final loss. Since it is equivalent in math to a negative Entropy loss term ($KL(q(z|x), Uniform(z))=\sum_zq(z|x)\log \big(q(z|x)*|Z|\big)=-Entropy + \log|Z|$), it is not easy to understand why this is useful intuitively. After all, the temperature $\tau$ can also take similar effects but is not very influential for performance in our experiments.

---

> > ### Comment · Reviewer_barm · 2021-09-01
> > **Reply**
> >
> > Thank you for providing additional examples and results on the FID-reranking and on the loss curves for the VQVAE variants. I also appreciate providing the inference time and estimated training costs.
> >
> > Let me comment on two points:
> >
> > (i), "Second, we think the reviewer gives a good comparison method in the ideal situation, but unfortunately our tokenizer is trained on only part of the dataset and cannot represent the data distribution in stage 2."
> >
> > I was referring to a possible evaluation and comparison of the two different tokenizers (i.e., VQVAE from CogView and dVAE from DALL-E). It would be sufficient to collect/find some test data for training the two tokenizers and then calculate the reconstruction metrics (e.g., LPIPS, PSNR, Reconstruction-FID) using this test data.
> >
> > (ii): "VQGAN or most GAN-based methods can only work on a single domain".
> > There is a publicly available model trained on the very diverse OpenImages dataset (https://github.com/CompVis/taming-transformers) using the same tokenizer parameters (codebook dimension of 8192, H=W=32).

---

> > > ### Author Response · Authors · 2021-09-01
> > > **Reply**
> > >
> > > Thank you for your comments. We have already noticed the released VQGAN, and through ablation study, we find that the essential point is that the released model uses "diffusion" module with more parameters and attention. The previous observation is not accurate and we will update it.
> > > Thank you again for your careful reviews.

---

### Official Review · Reviewer_RfQ5 · 2021-07-18

**Rating:** 5
**Confidence:** 4

**Summary:**

The paper proposes a new text-image pretrained model based on Transformer with VQVAE tokeniser, which can be implemented on different downstream tasks to achieve an appropriate performance. Also, two techniques called PB-relaxation and Sandwich-LN are proposed to reduce overflow in forwarding and to stabilise the training.

**Limitations And Societal Impact:**

Limitation:
1. Are authors going to make the collected dataset public? To some extent, the good performance might be benefited from the large-size training dataset.

2. The proposed method is similar as Dall-E as both rely on the GPT and VQVAE. It might be better if authors can show more discussion and comparison between proposed method and the Dall-E.





**Main Review:**

Originality: The proposed method is built on GPT with VQVAE tokeniser.

Quality: The submission is technically sound, claims are well supported, but authors do not discussed strengths and weaknesses of the proposed method.

Clarity: The paper is well organised and easily followed.

Significance: The results look good, which achieves good performance on different downstream tasks.

**Time Spent Reviewing:**

8

---

> ### Author Response · Authors · 2021-08-08
> **Authors' response**
>
> Thank you for your valuable comments, we will explain your concerns point by point.
>
> * Limitation 1: "Are authors going to make the collected dataset public? To some extent, the good performance might be benefited from the large-size training dataset "
>
>   We would like to, but the copyright of crawled images does not belong to us. We are planing to release the website names, and keywords extracted from the knowledge graph if accepted. The dataset of DALL-E is 5x larger than that of CogView, so the difference in algorithms (stabilization and finetuning) may be more important reasons for the improvements.
>
> * Limitation 2: " It might be better if authors can show more discussion and comparison between proposed method and the Dall-E."
>
>   DALL-E is a close-source and (actually) concurrent work of us, so it is not easy to compare with it in details. We have already compared CogView with DALL-E as much as we can, including concluding the contributions over DALL-E (Line 58-72), difference in tokenization techniques (Line 125-131), loss function(Line 140-141), stabilization techniques (Line 156-168), finetuning (Line 215, **DALL-E doesn't finetune**) and performance (Line 267-276 and Table 1).
>
>   I understand that the reviewer wants to a more detailed ablation studies with DALL-E on the different techniques, but it is impossible for us to compare with a close-source project in every details.  It should not be attributed to us. **This also exhibits the importance of open-sourcing, which is also an important contribution of CogView.**
>
> If our answers above solve your concerns, could you increase your rating?

---

### Official Review · Reviewer_ccEj · 2021-07-18

**Rating:** 6
**Confidence:** 5

**Summary:**

The paper proposes CogView - a cross-modal Transformer with VQ-VAE as the tokenizer. The model is primarily designed for text-to-image generation but also shows promising results on various downstream tasks.
The paper also provides many insights on the effective training tricks (including PB-Relax and Sandwich-LN) to stabilize the training of text-to-image transformers. The experimental results show that CogView outperforms the recent state-of-the-art approach DALL-E in the zero-shot settings.

**Ethics Review Area:**

["I don’t know"]

**Limitations And Societal Impact:**

The proposed CogView approach may be used for malicious purposes to generate fake images. That's a potential negative societal impact.

**Main Review:**

Strengths:
1) The paper is generally well-written and easy to follow.
2) The paper introduces several effective methods to stabilize the training for text-to-image generation.
3) The proposed CogView can be further extended to more tasks via fine-tuning: style learning, super-resolution, self-reranking.
4) The code and pre-trained model is open-source, which is the first for large text-to-image transformer, many applications will benefit from it.

Weakness:
1) The technical novelty of the paper is limited, its model is almost the same as DALL-E. The main contribution lies in the training strategy rather than in the techniques.
2) Training data is one possible reason why CogView beats DALL-E. Although Appendix A provides some descriptions of data collection. However, there are still not enough details to replicate the high-quality text-image pairs from the current descriptions.
3) It is hard to tell whether the improvement comes from the high-quality training data or the training tricks.
4) Self-reranking after fine-tuning is just another CLIP, it is still post-selection and does not get rid of CLIP.

Questions to the authors:
1) CogView used different data to train the VQ-VAE and text-to-image Transformer. DALL-E also released their own trained VQ-VAE, how will the results be like if using the same VQ-VAE and compare the generation performance between CogView and VQ-VAE?
2) How do you evaluate the ratio of training data and training tricks in the success of CogView?
3) In Table 1, how did you replicate the performance of DALL-E, why CapS didn’t apply to DALL-E? Or do you just copy the number of DALL-E on MSCOCO?


**Time Spent Reviewing:**

3 hours

---

> ### Author Response · Authors · 2021-08-08
> **Authors' response**
>
> Thank you for your valuable comments, we will explain your concerns point by point.
>
> * Weakness 1, "The technical novelty of the paper is limited over DALL-E."
>
>   In the introduction section, we list 4 main contributions over DALL-E, which are also the main contents of this paper. Here, we will stress **why the 4 contributions are important.**
>
>   * Contribution 1: New transformer layer backbone Sandwich-LN and PB-Relax. Although seeming to be an add-on at the first glance,  to stabilize of FP16 pertraining is the core and most challenging part in training text-to-image generation models, which is both argreed by CogView and  DALL-E (section 2.4 paragraph 1). **Firstly**, training large models usually relies on deepspeed ZeRO (Line 157), which only supports FP16. But the ordinary Pre-LN Transformer under FP16 cannot be trained at all at this task, which will quickly go into unskipable overflow (Line 158-160). **Secondly**, Sandwich-LN + PB-Relax is much more simple and effective than the specific mix-precision system of DALL-E (not very detailed and close-source), so our solution is the only public way to train a large text-to-image model currently.  **Moreover**, as discussed in the paper, the new normalization backbone additional stabilizes 10B NLP models and 64-layer toy model, meaning that its potential is beyond this task.
>   * Contribution 2: First to propose finetuning strategies for text-to-image generation. DALL-E leaves finetuning for future work (section 3.1 paragraph 5), while CogView proposes a series finetuning methods for different finetuning tasks. Among them, **super-resolution greatly increase the quality of text-to-image generation with very few additional computational resource for training.** And caption model can be used for post-selection without training an expensive CLIP. The finetuning only need 1 DGX machine for 1 day, but can benefit various task. This is also a very important contribution.
>   * Contribution 3: New evaluation metrics for text-to-image generation. We find the drawback of previous metrics according to human evaluation, and propose a new metrics CapS, which is based on CogView Caption model and corresponds the human evaluation well. This will prevent the development of this topic from being hindered by improper metrics in the future.
>   * Contribution 4. Better performance and open-sourcing. Although both   target at text-to-image generation, CogView (base) achieves better scores, and CogView-finetuned (super-resolution) performs even much better. DALL-E is close-source, which makes us hard to perform ablation study to find which part is the main factor for Cogview to outperform it. So we really need an open-source pretrained text-to-image model to benefit the development of this topic -- which is CogView. This is a big contribution.
>   * Other contributions: (1) The three-region sparse attention with 2.5x faster and 0.6x memory without hurting performance in Appendix B. (2) A thorough investigation and visualization of the attention in the text-to-image generation tasks in Appendix C. (3) A useful summary and comparison of VQVAE tokenizer in the section 2.2. (4) the usage of VQGAN in text-to-image generation for the first time in the section 3.4. They are not the main points of this paper but are also very helpful to the researchers in this area.
>
> * Weakness 2: "there are still not enough details to replicate the high-quality text-image pairs from the current descriptions (for dataset)".
>
>   I guess that your "details" are the exact website urls, the keyword list and the codes to extract the keywords from knowledge graphs. There are about 50 websites, which are manually selected by the authors,  and we will upload all the mentioned things to our repo if accepted.
>
> * Weakness 3: "It is hard to tell whether the improvement comes from the high-quality training data or the training tricks."
>
>   We have already compared CogView with DALL-E as much as possible, including the techniques in each part, the results on MS COCO, etc. DALL-E is not open-source, so we cannot compare them in every details to perform ablation studies --- **but this is a drawback of DALL-E, not a weakness of CogView**. This also exhibits the importance of open-sourcing, which is a contribution of CogView.
>
> * Weakness 4: "Self-reranking after fine-tuning is just another CLIP, it is still post-selection and does not get rid of CLIP"
>
>   **This is an advantage, not a weakness.** The difference is that fine-tuning saves a lot of resources. Training a model like CLIP consumes a lot of computation resource (at least $100,000 by our estimation) and data. As mentioned in the paper, finetuning CogView for self-ranking only need 1 machine for 1 day, which is a big advantage.
>
>   (To avoid ambiguity, could we use "ranking" instead of CLIP to refer the function in the discussion later? CLIP is a very good text-image ranking work, but is not equvalent to the "ranking" function.)
>
> * Question 1:
>
>   This work is actually concurrent with DALL-E. When the VQVAE of DALL-E released, the pretraining of CogView was almostly finished. Since each pretraining comsumes a lot of money (about $300,000), it is not possible (and have no reason) to change our tokenizer to re-train. The VQVAE will prefer the texture of trained data (maybe more rendered data in the dataset of DALL-E), but I don't think there will be big difference.
>
> * Question 2:
>
>   We cannot know the exact reason why CogView (w/o super-resolution) performs better than DALL-E because DALL-E is close-source. We are sorry that there is a typo in Line 277 to miss "Besides the different normalization achitecture in the Transformer Layer," before "We guess the main reason is that our data ... mainly photos."  What we actually want to express is that, even if the data of CogView only 1/5 of that of DALL-E, as researchers, we cannot directly owe the success totally to our new techniques instead of data, because the data distribution is also important. As we find CogView can also generate good cartoons with proper prompt after submission, we will rewrite this paragraph later.
>
> * Question 3:
>
>   Yes, we use the results from their paper." Statistics about DALL-E are extracted from their figures", which is written in the Caption of Table 1.
>
> If our answers above solve your concerns, could you increase your rating?

---

### Decision · Program_Chairs · 2021-09-27

**Decision:**

Accept (Poster)

**Comment:**

This work presents a text-image generation model using VQ-VAE to discretize images, which is then followed by a LM on text + image tokens. There was a debate whether this work is concurrent to or a follow-up of DALL-E (blog post on Jan 5, 2021 and arXiv on Feb 24, 2021, more than two months before NeuRIPS deadline) given similarities among the two approaches. When viewed as a follow-up work, I found that there is a reasonable amount of contributions (including technical ones) in this paper, e.g., the finetuning ideas for super-resolution (which greatly enhances the quality and is actually a neat idea) and self-ranking (without having to train a full CLIP model). The detailed findings of using nearest neighbors in VQ-VAE, setting equal weight to the LM loss for texts, etc., together with tricks for stabilizing training seem informative and valuable to the community. In addition, the authors promised to open-source this work. Because of these reasons, I recommend Accept.